# Uncertainty and Sensitivity of the Feature Selective Validation (FSV) Method

Jacopo Bongiorno [1] and Andrea Mariscotti [2,*]

1 Independent Researcher, 16043 Chiavari, Italy
2 Department of Electrical, Electronics and Telecommunication Engineering and Naval Architecture (DITEN), University of Genova, 16145 Genova, Italy
* Correspondence: andrea.mariscotti@unige.it

**Abstract:** The FSV method is a recognized validation tool that initially assesses the similarity between data sets for electromagnetic measurements and models. Its use may be extended to many problems and applications, and in particular, with relation to electrical systems, but it should be characterized in terms of its uncertainty, as for measurement tools. To this aim, the Guide to the Expression of Uncertainty in Measurement (GUM) is applied for the propagation of uncertainty from the experimental data to the Feature Selective Validation (FSV) quantities, using Monte Carlo analysis as confirmation, which ultimately remains the most reliable approach to determine the propagation of uncertainty, given the significant FSV non-linearity. Such non-linearity in fact compromises the accuracy of the Taylor approximation supporting the use of first-order derivatives (and derivative terms in general). MCM results are instead more stable and show sensitivity vs. input data uncertainty in the order of 10 to 100, highly depending on the local data samples value. To this aim, normalized sensitivity coefficients are also reported, in an attempt to attenuate the scale effects, redistributing the observed sensitivity values that, however, remain in the said range, up to about 100.

**Keywords:** electromagnetic modeling; FSV method; IEEE standards; simulation; uncertainty; validation; verification

## 1. Introduction

Numerical models are widely used to assess performance, reliability, and safety in various conditions and configurations for many systems and applications. Models can replace the physical system [1] for several reasons: the system may be still in the design phase, or it may be inaccessible or cannot be modified to perform the required tests; for example, when aiming at testing exceptional, and possibly critical or dangerous, conditions, as well as when the model represents a cheaper and faster solution, especially in the case of complex or new fabrication technologies [2,3]. Models were used to this aim, for example, when integrated circuits at the very beginning had extremely large fabrication costs and uncertain fabrication times [2], or when expensive materials are used, such as for superconductive coils [3]. Similarly, direct voltage control for complex scenarios of interconnected renewables, and wind parks in particular, is fully tested using a hardware-in-the-loop model in highly severe dynamic conditions, providing evidence of real-world applicability and design margins beyond the availability of a single instance of installation [4]. In this particular case, the proposed digital twin validation is achieved by comparison with another trusted model running on a different simulator, in line with the approach outlined in [5]. However, in the specific case of [4], the discussed validation results are only qualitative, displaying overlapped curves, without a quantitative estimation of the degree of similarity; thus, as a matter of fact, they leave the indeterminacy of the accuracy of the proposed model. In addition, a thorough model validation was carried out with multiple criteria, including FSV, in some different electronic and electrical applications: load forecasting by means of wavelets [6], the impact of batteries on DC grid impedance and

the transmission efficiency of power line communication signals [7], and the demonstration of efficient shielding of an inductive power transfer system [8].

When comparing simulated and experimental data (or those originating from a trusted model), several characteristics denoting the similarity can be collected and processed to formulate the necessary judgment: curve shapes, peaks, and slopes are all elements that capture the attention of the expert, who will put in the background other features that are often caused by noise, outliers, and other artifacts.

Several validation methods have been proposed in various fields of science; for example, those applied to economics, crystallography, and electromagnetics, and they have been reviewed, discussed, and quantitatively assessed and confronted for some test cases [9]. The relatively recent Feature Selective Validation (FSV) method [10] is a complex and complete validation tool with wide applicability to several fields of engineering, with particular emphasis on electronic and electrical engineering: pure EMC applications were presented in 2010 from studying reverberating chambers [11], and near- and far field radiation from heatsinks [12]; a thorough validation of a railway line model against experimental data is discussed in [13], with considerations on the influence of second-order system elements; counterfeiting detection via electromagnetic fingerprint is instead evaluated in [14]; finally, the modeling of a high voltage direct current system is discussed in [15]. Older examples can be found, especially in the field of microelectronics, when models of semiconductor devices began to appear at different levels of complexity [1], and are now widely available resources in many desktop circuit simulators.

A validation tool should be considered as an instrument that performs a quantitative assessment of an existing instrument (the model); in other words, a calibration. This is achieved by comparing a set of simulation outputs with a set of reference data, which may be experimental measurements or other simulation results, possibly provided by a trusted (validated) model or simulator. Such a validation instrument should be characterized in terms of uncertainty, so as to provide a final statement of confidence intervals for the observed accuracy of the validated model. For the purpose of verifying the FSV method itself, it can be fed with experimental or simulation results only, providing data curves with a controlled amount of diversity (e.g., slightly adjusting one parameter of the experiment, or simply repeating experiments under identical conditions for the utmost similarity, except for noise and repeatability issues).

The FSV tool is particularly useful when a simple metric that measures the distance (or error) between the simulation output and the reference data is not sufficient. The reason is that simple metrics, based, for instance, on Euclidean distance (known also as L2 norm) or absolute difference (known also as L∞ norm) calculated with a bin-to-bin correspondence are not able to capture the similarity of shape and slope, and they miss the similarity between slightly shifted and stretched curves (as is commonplace when there are slight deviations of parameters of frequency responses).

From a metrological standpoint, the resulting uncertainty of the validated model will never be better than the uncertainty of the validation instrument, and efforts should be undertaken to minimize the uncertainty of the reference cases (experimental data) and their propagation through the validation instrument itself (the FSV method, in our case).

Uncertainty evaluation for FSV is not completely new [16,17], but these approaches were semi-quantitative and uncertainty propagation was never considered. In particular, in [16], the term uncertainty is erroneously used as an equivalent of the error between data points, and the analysis is based on the interpretation of the crossing points of curves [17]. Focuses that are made instead on the histogram that is the very final result of the FSV method then provide an interesting analysis based on fuzzy set theory, but they neglect the behavior of the intermediate quantities amplitude difference measure (*ADM*) and feature difference measure (*FDM*), which will be discussed in Section 2.

Considering FSV as a measuring gauge to assess the performance of a model against reference data, the GUM (Guide to the Expression of Uncertainty in Measurement) [18] should be used for its characterization. In the following, the law of propagation of uncer-

tainty (LPU) through the algorithm is calculated by determining the sensitivity coefficients as first-order derivatives, then by looking for a confirmation using Monte Carlo analysis, which instead provides a more general method to estimate dispersion and confidence interval. The FSV implementation used for the evaluation takes into consideration the findings in [19], as summarized in Section 2.

The work is then structured as follows. Section 2 briefly presents the FSV algorithm implementation, making reference to previous literature to document the details. Section 3 simply exposes the Type B approach to the LPU determination through first-order derivatives as sensitivity coefficients. Section 4 reports the calculated derivatives and sensitivity coefficients of the FSV formulation, which is a novel analysis approach. The Monte Carlo method (MCM), as a more generally applicable method (as indicated by GUM), is then introduced in Section 5, clarifying the simulation conditions. The results for selected test cases are then reported and discussed in Section 6, providing quantitative results for the reliability of the Type B approach through derivatives, the sensitivity values calculated using the MCM approach and their dependency on data values, and ways to provide sensitivity information for general use.

## 2. Implementation of the FSV Method

The problem of the validation of a model may be stated as the quantitative assessment of the similarity between the model output *o* and the experimental data *m* (which we said may also be from another model or theoretical results). The discussion of FSV implementation (IEEE Std. 1597.1 [10]) and its possible pitfalls and unclear points, with consequences for variability and systematic errors, can be found in [19]. Only the main elements for supporting the estimate of uncertainty and its discussion are reported below: the approach and equations are taken from the IEEE Std. 1597.1, besides other references where explicitly indicated.

The FSV assessment can be decomposed in three phases:

- First, the original data sets are preprocessed to extract the "*dc*", low frequency ("*lo*") and high frequency ("*hi*") portions, obtained via first a Fourier transform (FFT), followed by Inverse FFT, after cutting the frequency intervals and smoothing the data;
- The three sub-intervals are fed to the second phase, where the ADM and FDM quantities are calculated;
- Their root-mean-square combination is the Global Difference Measure (GDM) and represents the last phase.

For the presentation of results, a useful technique is the "Grade and Spread" diagram [20–22]: the spread term measures the spread of the distribution, and the grade term is similar to skewness. As a note, for an assessment of the results, skewness and kurtosis will be used, as the shape of the distribution has an influence on the confidence interval determination.

When calculating the FFT, a zero padding of vectors should be avoided, using vectors that are as close as possible to their original length and possibly FFT, with prime-number factorization.

For the padding of points missing after the calculation of the derivative, the two methods "taper" and "fcb" show very similar results, which are most important for the categories of Excellent and Very Good, where a minimum of dispersion is desirable: the "taper" method is implemented in the following. To clarify, "taper" and "fcb" are defined in [19] as progressively reducing the distance between the two samples used to calculate the derivative (until the first and last points are reached) for "taper", or using a forward, center, and backward difference calculation for "fcb".

### 2.1. Vectors Preparation

Identified as the first step, the original data vector *x* in domain d is decomposed into *dc*, *lo*, and *hi* vectors, proceeding through the transformed domain D. The separation is performed with a DFT (Discrete Fourier Transform)/IDFT (Inverse DFT) pair [10], identifying

the three intervals *DC*, *LO*, and *HI* in domain D based on the calculation of the $I_b$ index: the first five data samples, the samples up to index $I_b$, and then that remaining beyond $I_b$.

$$X = DFT\{x\} \; DC' = X[0, 1, \dots, 4]$$
$$LO' = X[5, 6, \dots, I_b] \; HI' = X[I_b + 1, \; I_b + 2, \dots, N - 1] \tag{1}$$

The amount of applied zero padding, *zp*, is the minimum for bringing the vectors *DC* (used only for ODM), *LO*, and *HI* to the same length, so as to combine then their anti-transformed versions for the calculation of indexes.

$$DC = zp(DC') \; LO = zp(LO') \; HI = zp(HI') \tag{2}$$

$$dc = \mathrm{IDFT}\{DC\} \; lo = \mathrm{IDFT}\{LO\} \; hi = \mathrm{IDFT}\{HI\} \tag{3}$$

The determination of $I_b$ is based on the known 40% threshold that is applied to the area of the curve (that means $I_b$ indicates the index at which the low-frequency portion of data points amounts to 40% of the total).

$$I_b : \sum_{i=0}^{I_b} |X[i]| = 0.4 \sum_{i=0}^{N-1} |X[i]| \tag{4}$$

Vector separation across $I_b$ is implemented with a linear tapering across $N_b$ samples; then, the IDFT for each of them is calculated to return to the original domain, as indicated in (3).

### 2.2. Calculation of FSV Indexes

To complete steps two and three, the *ADM*, *FDM*, and *GDM* indexes are calculated, as shown below. Equations (5)–(12) are taken from the IEEE Std. 1597.1 [10], and are slightly reshaped for clarity. In particular, the numeric coefficients of the FDM expressions are provided in the IEEE Std. 1597.1, without an explanation, but they are supposed to come from the number of samples used in the derivative operators described in the standard itself.

$$ADM_i = \frac{\left| |o_{lo,i}| - |m_{lo,i}| \right|}{\frac{1}{N}\sum_{j=0}^{N-1} |o_{lo,j}| + |m_{lo,j}|} + ODM_i \exp^{ODM_i} \tag{5}$$

$$ODM_i = \frac{\chi_i}{\delta_i} = \frac{\left| |o_{dc,i}| - |m_{dc,i}| \right|}{\frac{1}{N}\sum_{j=0}^{N-1} |o_{dc,j}| + |m_{dc,j}|} \tag{6}$$

$$FDM_i^1 = \frac{|o'_{lo,i}| - |m'_{lo,i}|}{\frac{2}{N}\sum_{j=0}^{N-1} |o'_{lo,j}| + |m'_{lo,j}|} \tag{7}$$

$$FDM_i^2 = \frac{|o'_{hi,i}| - |m'_{hi,i}|}{\frac{6}{N}\sum_{j=0}^{N-1} |o'_{hi,j}| + |m'_{hi,j}|} \tag{8}$$

$$FDM_i^3 = \frac{|o''_{hi,i}| - |m''_{hi,i}|}{\frac{7.2}{N}\sum_{j=0}^{N-1} |o''_{hi,j}| + |m''_{hi,j}|} \tag{9}$$

$$FDM_i = 2 \left| FDM_i^1 + FDM_i^2 + FDM_i^3 \right| \tag{10}$$

The prime and double prime indicate the first and second derivatives, and the subscripts indicate the considered vector portion ("*lo*" or "*hi*"). Derivatives in *FDM$_i$* are calculated as a difference, using an index interval that is ±2 for "lo" and ±3 for "hi", but without dividing by the differential of the independent variable, which would make these terms difference quotients. This may be identified as a little pitfall, and will be considered again in Section 3, when the propagation of uncertainty is calculated.

*GDM* is calculated using the original definition [10], which is the root summed square of the $ADM_i$ and $FDM_i$ terms, as shown in (11) and (12):

$$GDM_i = \sqrt{(ADM_i)^2 + (FDM_i)^2} \tag{11}$$

$$GDM = \sum_{i=0}^{N-1} GDM_i \tag{12}$$

### 2.3. Visualization and Interpretation

The IEEE Std. 1597.1 requires classification of the FSV *GDM* values with an interpretation scale, as in Table 1.

**Table 1.** Interpretation scale for *GDM*.

| Lower Bound | Upper Bound | Quality Descriptor |
|---|---|---|
| 0.0 | 0.1 | Excellent (E) |
| 0.1 | 0.2 | Very Good (VG) |
| 0.2 | 0.4 | Good (G) |
| 0.4 | 0.8 | Fair (F) |
| 0.8 | 1.6 | Poor (P) |
| 1.6 | $+\infty$ | Very Poor (VP) |

## 3. Propagation of Uncertainty

When speaking of model uncertainty, it is assumed that systematic errors have been compensated for or removed, and that the remaining error is random and may be described by random variables and their statistics. Examples of systematic errors for the FSV algorithm itself are those considered in [19], due to ambiguity in some points of the standard and implementation issues. Of course, other systematic errors may affect the measured data, and also in this case, it is supposed that they have been removed, or that they are negligible.

The quantities are then characterized by uncertainty regarding their estimate, which is best expressed as the dispersion of their probability density function (PDF), either derived from the observed distribution (Type A approach [18]), or based on the available information (or some degree of belief) of the possible sources of variability (Type B approach [18]).

For the propagation of uncertainty, a model is necessary; i.e., an algebraic relationship $F$ between the input quantities $X_i$ and the output $Z$: $Z = F(X_i)$. In general, the propagation of uncertainty between two quantities $p$ and $q$ is based on sensitivity coefficients $S_p(q)$, which indicate that uncertainty propagates from $q$ (the independent variable) to $p$, and they are determined as first-order derivatives between the two quantities of interest.

Using first-order derivatives, a linearization by Taylor expansion is implicit around a point $\mu$, defined by the expectation of $X_i$ (or mean) $\mu_i = E(X_i)$, as shown in (13).

$$z = F(\mu) + \sum_{i=1}^{n} \left( \frac{\partial F}{\partial x_i} \right)_\mu (x_i - \mu_i) \tag{13}$$

The first term is the expectation of the output, and the second term determines the uncertainty of $z$. Two important limitations arise:

1.  To apply (13), the non-linearity of F must be negligible (GUM Clause 5.1.2 [18]); otherwise, the linear truncation in the Taylor expansion could lead to misleading results (we will see that this is often the case due to the significant non-linearity in the FSV method);
2.  The assumption of normality of $Z$ derived from the application of the Central Limit Theorem (GUM Clause 5.1.2 [18]) should be considered with caution; this will be

verified indirectly by calculating the skewness and kurtosis of the MCM output, but not by thoroughly assessing the normality of distribution.

Related to point 1, a key element that needs verification is the use of absolute values for both the data vectors and the FSV derivative: such non-linearity is variable and depends on the type of data fed to FSV, and can thus be assessed accurately only on a case-by-case basis, which thus undermines the generality of the approach based on derivatives.

The so obtained closed-form uncertainty expressions are then validated via MCM, which can similarly be applied to cases where the formal calculation of derivatives is impossible or too complex. MCM is by all means a valid and effective tool in general, and for situations in which it is difficult to apply the GUM uncertainty framework, or where its conditions are not fulfilled. Taking from [23]:

1.  The contributory uncertainties are not of the same magnitude: it is evident that *ADM* and *FDM* respond differently to different types of input data (e.g., containing wide slopes or narrow zig-zags, or some amount of noise);
2.  The calculation of the partial derivatives is affected by heavy approximations, or is even not possible, as it turns out when considering non-linearity;
3.  The probability distributions are not the assumed Gaussian or Student *t*, which was verified, at least for the *GDM*, in [20];
4.  As often occurs, PDFs are not symmetric [20]; or
5.  The output quantities are characterized by a significant standard uncertainty in the order of their magnitude; this last statement is added for completeness, but it is in general not relevant, as long as input data have a good measurement quality, with uncertainty in the order of some % or less.

## 4. Law of Propagation of Uncertainty and First-Order Derivatives

This section reports the core of the problem, i.e., verifying whether an approach to uncertainty propagation based on first-order derivatives is viable and accurate.

The determination of the propagation of uncertainty is achieved by the calculation of the partial derivatives of relevant FSV indexes, namely $ADM_i$ and $FDM_i$, with respect to one of the input vectors (the $m$ vector is assumed without loss of generality), assuming the other vector $o$ is held fixed (the $m$ and $o$ vectors are, in reality, interchangeable). This approach is novel as it could not be found in the related literature.

First-order derivatives are calculated based on the assumption that all variables are uncorrelated, as is customary. Equation (14) repeats the $ADM_i$ expression, making explicit the terms that are then derived in (15) to (19) with respect to the applicable input data, namely $m_{lo,i}$ and $m_{dc,i}$. The derivative sensitivity coefficients for the three $FDM_i$ terms are calculated in (20) to (22), and then the overall derivatives with respect to the applicable input data ($m_{lo,i}$ and $m_{hi,i}$) are shown in (23) and (24).

$$ADM_i = \frac{\left|\, |o_{lo,i}| - |m_{lo,i}| \,\right|}{\frac{1}{N}\sum_{j=0}^{N-1} \left|o_{lo,j}\right| + \left|m_{lo,j}\right|} + ODM_i \exp^{ODM_i} = \frac{\alpha_i}{\beta_i} + \frac{\chi_i}{\delta_i}\exp\left(\frac{\chi_i}{\delta_i}\right) = \gamma_i + \theta_i e^{\theta_i} \tag{14}$$

$$S_{\alpha_i}(m_{lo,i}) = \frac{\partial \alpha_i}{\partial m_{lo,i}} = -\mathrm{sgn}\left(\,|o_{lo,i}| - |m_{lo,i}|\,\right)\mathrm{sgn}(m_{lo,i})\; S_{\beta_i}(m_{lo,i}) = \frac{\partial \beta_i}{\partial m_{lo,i}} = \frac{1}{2N}\mathrm{sgn}(m_{lo,i}) \tag{15}$$

$$S_{\chi_i}(m_{dc,i}) = \frac{\partial \chi_i}{\partial m_{dc,i}} = -\mathrm{sgn}\left(\,|o_{dc,i}| - |m_{dc,i}|\,\right)\mathrm{sgn}(m_{dc,i})\; S_{\delta_i}(m_{dc,i}) = \frac{\partial \delta_i}{\partial m_{dc,i}} = \frac{1}{2N}\mathrm{sgn}(m_{dc,i}) \tag{16}$$

$$S_{\gamma_i}(m_{lo,i}) = \frac{\partial(\alpha_i/\beta_i)}{\partial m_{lo,i}} = \frac{\alpha_i'\beta_i - \alpha_i\beta_i'}{\beta_i^2} = \frac{-\beta_i\mathrm{sgn}\left(\,|o_{lo,i}| - |m_{lo,i}|\,\right)\mathrm{sgn}(m_{lo,i}) - \frac{1}{2N}\left|\,|o_{lo,i}| - |m_{lo,i}|\,\right|\mathrm{sgn}(m_{lo,i})}{\beta_i^2} \tag{17}$$

$$S_{\theta_i}(m_{dc,i}) = \frac{\partial(\chi_i/\delta_i)}{\partial m_{dc,i}} = \frac{\chi_i'\delta_i - \chi_i\delta_i'}{\delta_i^2} = \frac{-\delta_i \text{sgn}\big(|o_{dc,i}| - |m_{dc,i}|\big)\text{sgn}(m_{dc,i}) - \frac{1}{2N}\big||o_{dc,i}| - |m_{dc,i}|\big|\text{sgn}(m_{dc,i})}{\delta_i^2} \quad (18)$$

$$S_{ADM_i}(m_{lo,i}) = S_{\gamma_i}(m_{lo,i}) \; S_{ADM_i}(m_{dc,i}) = \frac{(\chi_i'\delta_i - \chi_i\delta_i')}{\delta_i^2}\left(\frac{\chi_i}{\delta_i}+1\right)e^{\chi_i/\delta_i} \; S_{ADM_i} = S_{ADM_i}(m_{lo,i}) + S_{ADM_i}(m_{dc,i}) \quad (19)$$

$$S_{FDM_i^1}(m_{lo,i}) = \frac{\partial m_{lo,i}'}{\partial m_{lo,i}}\frac{\partial FDM_i^1}{\partial m_{lo,i}'} = m_{lo,i}'' \frac{-\text{sgn}(m_{lo,i}')\frac{2}{N}\sum_{j=0}^{N-1}\left|o_{lo,j}'\right| + \left|m_{lo,j}'\right| - \frac{2}{N}\text{sgn}(m_{lo,i}')\left(\left|o_{lo,i}'\right| - \left|m_{lo,i}'\right|\right)}{\left[\frac{2}{N}\sum_{j=0}^{N-1}\left|o_{lo,j}'\right| + \left|m_{lo,j}'\right|\right]^2} \quad (20)$$

$$S_{FDM_i^2}(m_{hi,i}) = \frac{\partial m_{hi,i}'}{\partial m_{hi,i}}\frac{\partial FDM_i^2}{\partial m_{hi,i}'} = m_{hi,i}'' \frac{-\text{sgn}(m_{hi,i}')\frac{6}{N}\sum_{j=0}^{N-1}\left|o_{hi,j}'\right| + \left|m_{hi,j}'\right| - \frac{6}{N}\text{sgn}(m_{hi,i}')\left(\left|o_{hi,i}'\right| - \left|m_{hi,i}'\right|\right)}{\left[\frac{6}{N}\sum_{j=0}^{N-1}\left|o_{hi,j}'\right| + \left|m_{hi,j}'\right|\right]^2} \quad (21)$$

$$S_{FDM_i^3}(m_{hi,i}) = \frac{\partial m_{hi,i}''}{\partial m_{hi,i}}\frac{\partial FDM_i^3}{\partial m_{hi,i}''} = m_{hi,i}''' \frac{-\text{sgn}(m_{hi,i}'')\frac{7.2}{N}\sum_{j=0}^{N-1}\left|\hat{o}_{hi,j}''\right| + \left|\hat{m}_{hi,j}''\right| - \frac{7.2}{N}\text{sgn}(\hat{m}_{hi,i}'')\left(\left|\hat{o}_{hi,i}''\right| - \left|\hat{m}_{hi,i}''\right|\right)}{\left[\frac{7.2}{N}\sum_{j=0}^{N-1}\left|\hat{o}_{hi,j}''\right| + \left|\hat{m}_{hi,j}''\right|\right]^2} \quad (22)$$

$$S_{FDM_i}(m_{lo,i}) = \frac{\partial FDM_i}{\partial FDM_i^1}\frac{\partial FDM_i^1}{\partial m_{lo,i}} = 2\text{sgn}(FDM_i^1)\; S_{FDM_i^1}(m_{lo,i}) \quad (23)$$

$$S_{FDM_i}(m_{hi,i}) = \frac{\partial FDM_i}{\partial FDM_i^2}\frac{\partial FDM_i^2}{\partial m_{lo,i}} + \frac{\partial FDM_i}{\partial FDM_i^3}\frac{\partial FDM_i^3}{\partial m_{lo,i}} = 2\text{sgn}(FDM_i^2)\; S_{FDM_i^2}(m_{hi,i}) + 2\text{sgn}(FDM_i^3)\; S_{FDM_i^3}(m_{hi,i}) \quad (24)$$

It is observed that the derivatives to use for the propagation of uncertainty are intended in the "usual" way of limiting of the difference quotient, while the derivatives indicated in the original *FDM* expressions (indicated by a hat ˆ) are calculated simply as differences.

We may spend a few words on the selected method for derivative calculation in FSV and the IEEE Std. 1597.1. The objective is, of course, a reduction in the impact of noisy or peaky data: the numerator of the difference quotient picks two data points farther away, resulting in a sort of masking of small local peaks. Better smoothing is achieved using higher-order methods for the calculation of the derivative, such as the five-point stencil:

$$f' = \frac{f(x-2dx) - 8f(x-dx) + 8f(x-dx) - f(x+2dx)}{12dx} \quad (25)$$

Here, the numerator depends on the curve characteristics and on the variability of the samples. Applying an independent random variation at each sample and reasoning on the expectation, the average of the numerator does not change, whatever the method of calculation of the derivative. The standard deviation, conversely, will be lower, passing from a classic two-point difference to more elaborated schemes, such as the 5-point stencil.

It is also worth noting that the *FDM_i* sensitivity coefficients contain a second-order ((20) and (21)) or a third-order (22) derivative of the input data, in principle suffering from noisy or peaky data. We will see that they suffer indeed from edge effects, but they are not substantially larger than the *ADM_i* sensitivity values.

Going back to the objective of this section, the calculated sensitivity coefficients (see (15) through (24)) may be used to derive the uncertainty of the output quantity *GDM* described by (11) and (12). We may assume that the elements of the input data vectors are independent. However, a small amount of correlation between the adjacent FSV quantities is due to the calculation of derivatives made by taking the difference of adjacent samples: this is too complicated to propagate, and is in any case a second-order aspect. For this

reason, the FSV quantities $ADM_i$ and $FDM_i$ are assumed to be independent for each *i*-th sample of the input vectors.

## 5. Monte Carlo Method and Uncertainty Estimation

In the MCM-based uncertainty analysis, calculations are performed repeatedly, each with differing input values, to obtain probability distributions of the target output. Input values are sampled from their respective distributions, either known or assumed [24,25]. Calculations consist of running the model subject to assessment, which in the present case is the FSV evaluation for a given test data set. MCM is thus inherently computationally expensive, particularly when running the model requires much computing time. For the specific data vectors of the considered test cases (amounting to some hundreds of samples, as clarified below in Section 6) and the number of iterations *M* determined below, the computational time for the MATLAB scripts with figure generation was about 4 h, running within a virtual machine using Windows 7 64 bit, 16 GB RAM, and four physical Intel i7 10875H cores with 2.3 GHz clocks.

For the MCM computational load, a minimization of the number of runs is a relevant point to consider [26]. Two elements in general can reduce the number of runs and the overall computational effort and time: sampling efficiency and convergence monitoring.

The concept of sampling efficiency can be best described as the number of simulation runs required to reach some level of accuracy. A well-known variance-reduction technique for MCM is Latin hypercube sampling [27], with improved performance over random sampling, with the specific objective of better space filling.

The ability then to halt the analysis when a sufficient level of accuracy has been reached equally allows for a reduction in the overall duration and calculation time: this is achieved by monitoring convergence, followed by the addition of successive trials if the stop criterion is not met. When only the mean of the desired target output is relevant, an estimate of the sample variance can be used as a stop criterion; conversely, when more complex aggregate results are the objective (distributions, percentiles, and dispersion itself, as in the present case), the batch statistics and the similarity of distributions can be applied instead. However, whereas for random sampling the addition of new trials is straightforward, optimized space sampling techniques are not so flexible, since the sampling scheme is generally predetermined.

For this reason, in this work, the analysis for the determination of the uncertainty and the sensitivity coefficients is carried out using a basic random sampling scheme and by keeping the MCM code simple.

Sensitivity coefficients are determined by dividing the sample dispersion of the output $\sigma_{Zi}$ (in our case, corresponding to $\sigma_{ADMi}$ or $\sigma_{FDMi}$) by the applied standard deviation $\sigma_n$ of the injected noise at each MCM run.

With random sampling and a priori determination of the trials, the number of trials *M* for a given coverage probability *p* can be determined based on GUM [18]:

$$M > \frac{10^4}{1 - p} \tag{26}$$

For *p* = 95%, the resulting minimum number of trials is $2 \times 10^5$ (selected for the tests carried out in the following).

The combined uncertainty *u* and the confidence interval *CI* can be estimated from the set of the output results $Z_i$ using the known multiplicative factor of the sample standard deviation of the set $\sigma_{Zi}$, assuming that it is approximately Gaussian $u = 1.96\ \sigma_{Zi}$ for the mentioned *p* = 95% level of confidence.

As the resulting PDFs may be asymmetric (a behavior common to almost all indexes, as discussed in [1,20]), higher-order moments are relevant as well. Skewness *sk* and kurtosis *ku* are important parameters of the distribution of the output results. The accuracy provided by MCM is inversely proportional to the number of trials *M*: for a normal distribution $sk = 0$ with a standard deviation of $\sqrt{(6/M)}$, and $ku = 3$ with a standard deviation of

$\sqrt{(24/M)}$, that for the selected $M$ value are both well below 1%, so that the $sk$ and $ku$ estimates are to be considered "exact".

To limit the effects of the intrinsic non-linearity implicit in the FSV method, the applied MCM perturbation is limited to $\sigma_n = 0.1\%$: the random noise applied to each data sample is obtained using a zero-mean normal distribution with a dispersion equal to 0.1% of the data sample value.

## 6. Comparison and Verification of the Approaches to the Propagation of Uncertainty

The results obtained with the two methods (sensitivity coefficients with first-order derivatives of Section 4 and MCM of Section 5) are now compared for a set of test cases where raw data are available and the problem details are well known.

There is a significant degree of variability in the derivatives and a general sensitivity to the data range, which obviously depends on the problem nature, with the data being, for instance, large impedance values (e.g., railway pantograph impedance values [13,28]), near-unity reflection coefficients (as provided by scattering parameters of cable harness, transmission lines, and connectors), or very small electric field values (as is common in electromagnetic emissions measurements caused by electronic devices, power converters, and electrical machinery [29–31]), to give some examples. As such behavior was observed in the preliminary tests, additional verification of the suitable data pre-conditioning and derived indexes, with more general validity, was carried out.

Data were taken from two real test cases:

- Test case 1: impedance curves of a $2 \times 25$ kV 50 Hz railway line (Italian high-speed line), with measurements carried out between 100 Hz and 20 kHz for the purpose of the validation of line models [13]; this data set contains vectors of $N = 201$ data samples.
- Test case 2: scattering $S_{11}$ reflection loss, related to the input impedance of a test fixture for avionic connectors under various mechanical stresses (unpublished results); measurements were carried out with a vector network analyzer between 100 MHz and 6 GHz; this data set contains vectors of $N = 418$ data samples.

It is worth underlining that a characteristic of several performance indexes was used for model validation: the use of absolute values for index calculation in the case of noise injection causes under-utilization of the Excellent sub-interval, thus negatively biasing the overall judgment towards poorer performance [9,20].

Test cases were selected, so as to carry out the verification using real data for different degrees of similarity between the curves, and thus, of different $ADM_i$ and $FDM_i$ values: FSV expressions are in fact non-linear, and the values of the derivatives depend on the position along the data vector. Different degrees of similarity between pairs of data vectors are available; in particular, in test case 2: impedance curves were taken by moving along the railway line by 500 m each time, as shown later in Section 6.1, and at positions towards the end, there is better overlapping for adjacent positions.

Selected test cases are briefly introduced in the following, before providing a comparison of analytically calculated first-order derivatives with MCM results.

As will be demonstrated, the sensitivity of $ADM_i$ and $FDM_i$ changes significantly for the data provided in linear scale and using a logarithmic compression (dB); for this reason, the results of $ADM_i$ and $FDM_i$ sensitivity are shown for both scenarios for each test case.

### 6.1. Test Case 1: Railway Line Impedance

The tests carried out on a $2 \times 25$ kV 50 Hz railway line in Italy [13] provide a large set of curves, with various degrees of differences, thus allowing a test of FSV characteristics under a range of conditions (see Figure 1). It may be observed that all curves are similar at low frequency, and then begin to separate at some kHz.

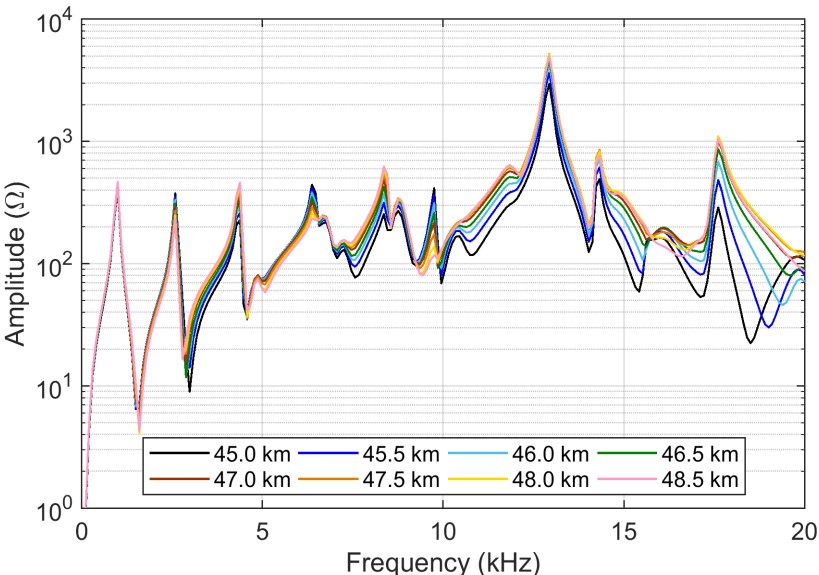

**Figure 1.** Line impedance curves of test case 1 [13] for positions along the line from 45.0 km to 48.5 km. Note the closer similarity of curves in the last positions beyond 47 km.

The curves displayed in Figure 1 are plotted with a logarithmic vertical axis, as is customary for this type of curves: this raises the point for if they should be provided to the FSV algorithm in a linear or log-compressed scale, the latter implying a dB representation. If in principle there is no reason to propend for one or another representation, we will see in the following a different behavior of the sensitivity coefficients.

6.1.1. Different Curves (45.0 and 45.5 km) with Linear and Logarithmic Scales

Two curves are selected with significant differences (as is visible in Figure 1, mostly a difference in amplitude, but also of resonance positions at a higher frequency) and providing data to FSV as they are, without logarithmic pre-processing. The tests consist of a preliminary check that analytically calculated $ADM_i$ and $FDM_i$ and those resulting from the average of the MCM simulation output correspond (a sanity check is made to spot out biasing of the MCM output), as shown in Figure 2.

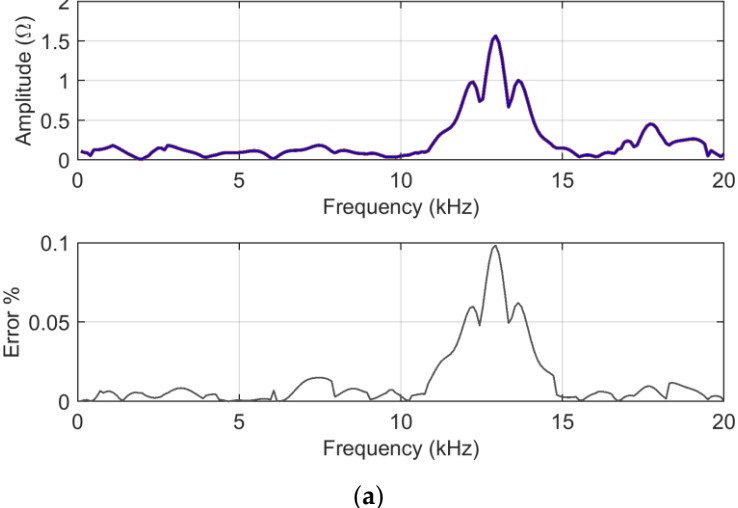

(**a**)

**Figure 2.** *Cont.*

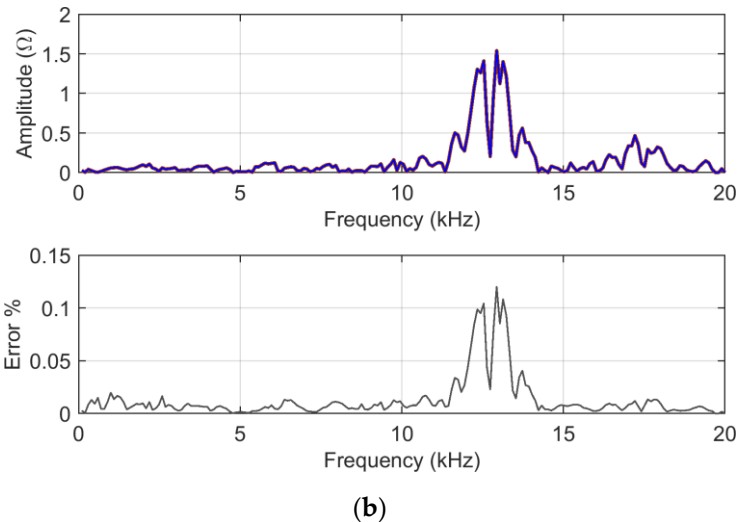

(**b**)

**Figure 2.** *Z* (45.0 km) and *Z* (45.5 km) input data in linear scale: (**a**) $ADM_i$ and (**b**) $FDM_i$ plots as provided by analytical calculation (blue) and by means of MCM output (brown); blue and brown curves are overlapped with underlying brown curve visible at the edges of the blue one. The relative error remains below 0.15% in all cases, demonstrating the reliability of MCM simulations and lack of biasing.

As shown in Figure 2, $ADM_i$ and $FDM_i$ capture the differences mainly between 10 and 15 kHz, and 16 and 20 kHz. In the first interval, $ADM_i$ and $FDM_i$ take higher values for a matter of scale, as explained below for logarithmic compression.

The relative error is calculated as the difference between the analytical FSV calculation and the mean of the MCM output normalized by their half sum.

For this first set of data, a thorough check is performed for the influence of the logarithmic compression of data, for which $ADM_i$ and $FDM_i$ are provided below in Figure 3, and also for the logarithmic case. The logarithmic compression has enhanced the differences between the curves occurring at lower amplitude, namely between 16 and 20 kHz, and now prevailing over those occurring between 10 and 15 kHz. It is evident, however, that the $ADM_i$ and $FDM_i$ values are smaller when logarithmic compression is used (see Figure 3), simply because the absolute value of the input data is smaller.

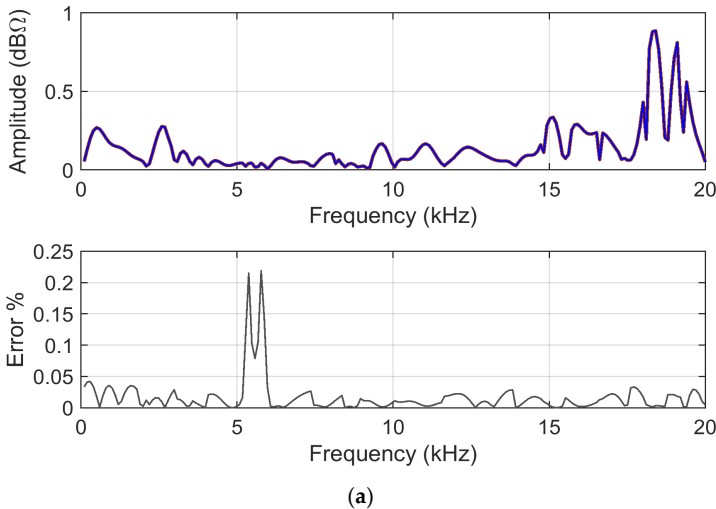

(**a**)

**Figure 3.** *Cont*.

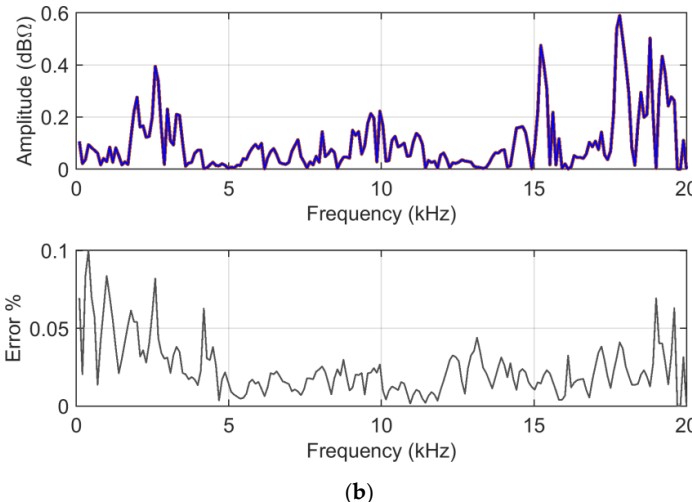

**(b)**

**Figure 3.** $Z$ (45.0 km) and $Z$ (45.5 km) input data in logarithmic scale: (**a**) $ADM_i$ and (**b**) $FDM_i$ plots as provided by analytical calculation (blue) and means of MCM output (brown); blue and brown curves are overlapped with underlying brown curve visible at the edges of the blue one. The relative error remains below 0.15% in all cases, demonstrating the reliability of MCM simulations and lack of biasing.

It may be concluded thus that logarithmic compression has helped form a more balanced assessment of curve similarity and diversity.

Sensitivity, as resulting from analytical calculations (the implementation of first-order terms of Section 4) and MCM simulations (as the standard deviation of the output), is shown in Figures 4 and 5.

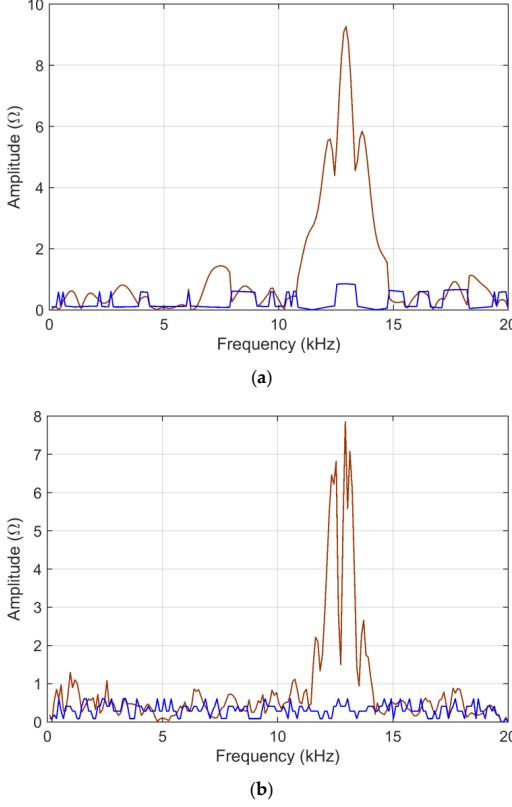

**(a)**

**(b)**

**Figure 4.** $Z$ (45.0 km) and $Z$ (45.5 km) input data in linear scale: (**a**) $ADM_i$ and (**b**) $FDM_i$ sensitivity curves, comparing analytical calculation (blue) and normalized standard deviation of MCM output (brown). The analytically calculated coefficients have been amplified by a factor of 100 for a matter of visibility.

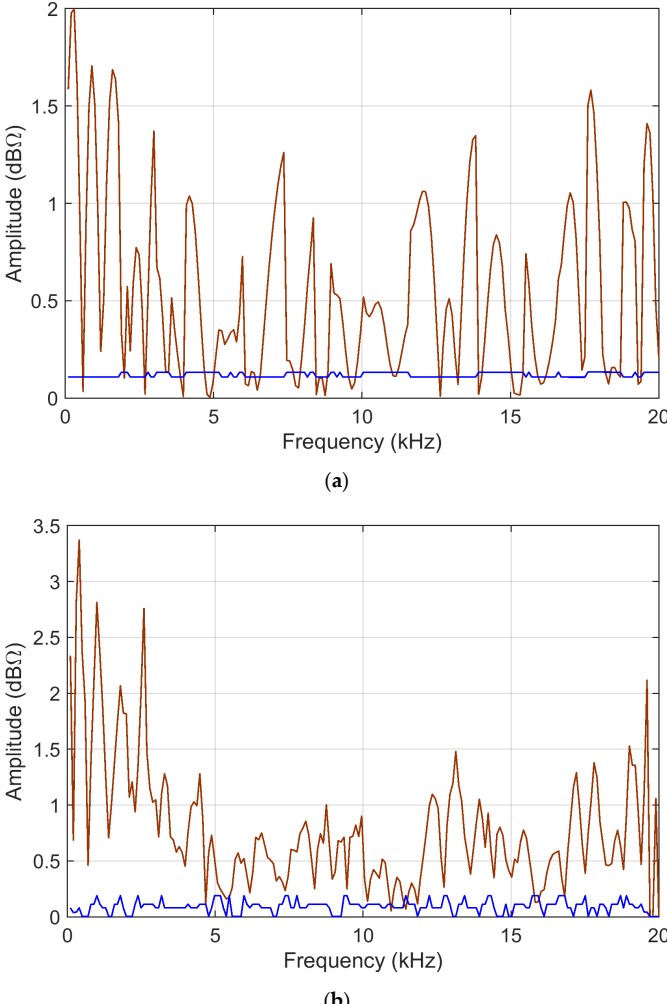

**Figure 5.** *Z* (45.0 km) and *Z* (45.5 km) input data in logarithmic scale: (**a**) $ADM_i$ and (**b**) $FDM_i$ sensitivity curves, comparing analytical first-order derivative calculation (blue) and normalized standard deviation of MCM output (brown).

As already observed, for the linear case, both the $ADM_i$ and $FDM_i$ quantities, and their sensitivity coefficients as well, have maximum values where the data is at the highest peak, at around 12 kHz. This confirms that all such quantities are not normalized, and suffer scale problems. By comparing Figure 4 (where the blue curve is amplified by a factor 100) and Figure 5, it is evident that providing data in the linear or logarithmic scale has a dramatic influence on the behavior of sensitivity: in this latter case, with logarithmic compression, the analytic and MCM curves are closer. This means that the problem with the logarithmic scale is formulated in a way that favors the analytic calculation of the sensitivity coefficients, and that in general, the sensitivity is lower, as confirmed by the MCM results, which are smaller by a factor of 4–5 and more uniform.

MCM output was checked for normality in a compact way, aiming at spotting out inconsistencies of sensitivity value distributions over the entire data vectors, where the output deviates from the expected normal distribution. To this aim, the skewness *sk* and kurtosis *ku* are plotted for $ADM_i$ and $FDM_i$ sensitivity (the latter being brought around 0 by subtracting the expected value of 3 for a Gaussian distribution). The results are shown in Figures 6 and 7.

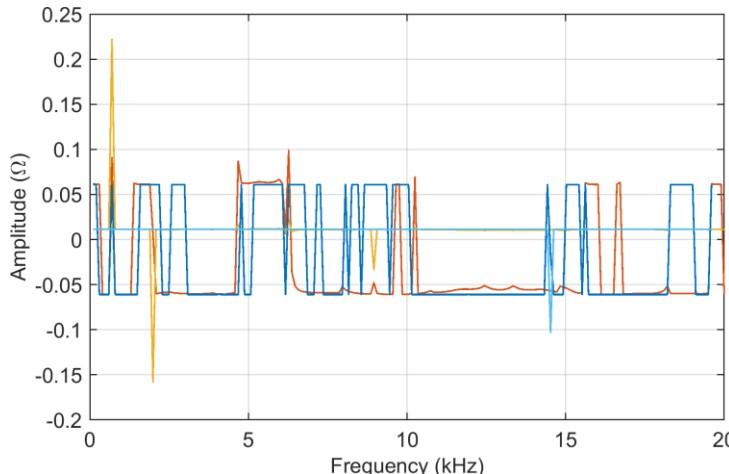

**Figure 6.** $Z$ (45.0 km) and $Z$ (45.5 km) input data in linear scale: skewness in dark color and kurtosis (offset around 0) in light color for $ADM_i$ (orange and yellow) and $FDM_i$ (blue and light blue).

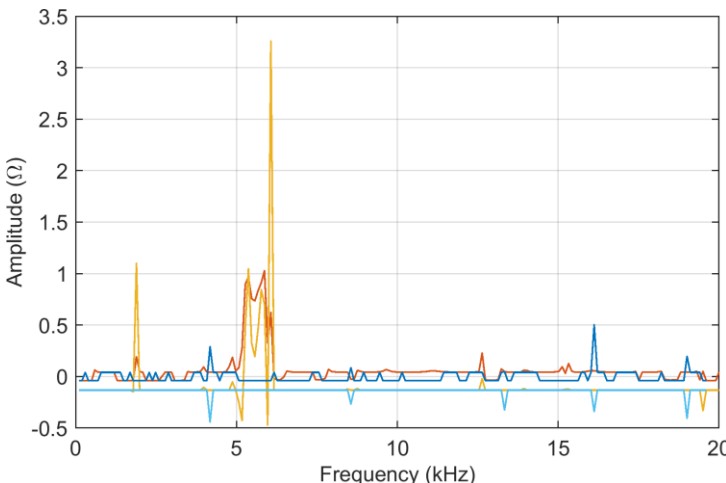

**Figure 7.** $Z$ (45.0 km) and $Z$ (45.5 km) input data in logarithmic scale: skewness in dark color and kurtosis (offset around 0) in light color for $ADM_i$ (orange $sk$ and yellow $ku$) and $FDM_i$ (blue $sk$ and light blue $ku$).

It may be said that $sk$ is limited to very small values, and thus, the output distribution is symmetric; in addition, except for a few points, $ku$ also indicates normality.

Last, having observed that the sensitivity coefficients $S_{ADMi}$ and $S_{FDMi}$ are influenced by the local magnitude of the data, the normalization of such coefficients is investigated in Section 6.3.

Another case is now verified, where the two curves are very similar, so that $ADM_i$ and $FDM_i$ should be smaller, in order to confirm at which extent sensitivity coefficients suffer from scale problems.

6.1.2. Similar Curves (47.5 and 48.0 km) with Linear and Logarithmic Scales

Two impedance curves are taken that are very close to each other, thus showing values of $ADM$ and $FDM$ that are at least 4–5 times smaller than those of the previous section (Section 6.1.1.), referring instead to more different impedance curves. $ADM_i$ and $FDM_i$ amplitudes for analytical and MCM calculations are shown in Figure 8. Figure 9 reports the $ADM_i$ and $FDM_i$ sensitivity curves. Figure 10 shows $sk$ and $ku$ for $ADM_i$ and $FDM_i$ calculated on the new alike impedance curves.

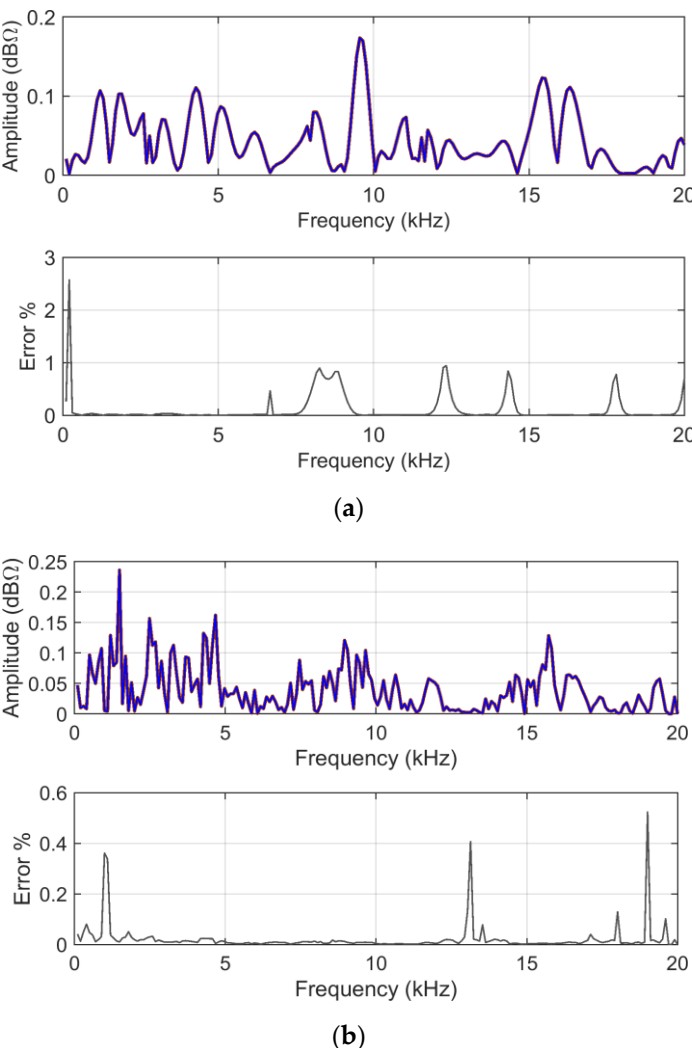

**Figure 8.** *Z* (47.5 km) and *Z* (48.0 km) input data in logarithmic scale: (**a**) *ADM$_i$* and (**b**) *FDM$_i$* plots as provided by analytical calculation (blue) and mean of MCM output (brown); blue and brown curves are overlapped with underlying brown curve visible at the edges of the blue one. The relative error has increased to a fraction of % or a few % at some points, where the *ADM$_i$* or *FDM$_i$* values are the lowest; on average, correspondence between analytical and MCM curves is quite good.

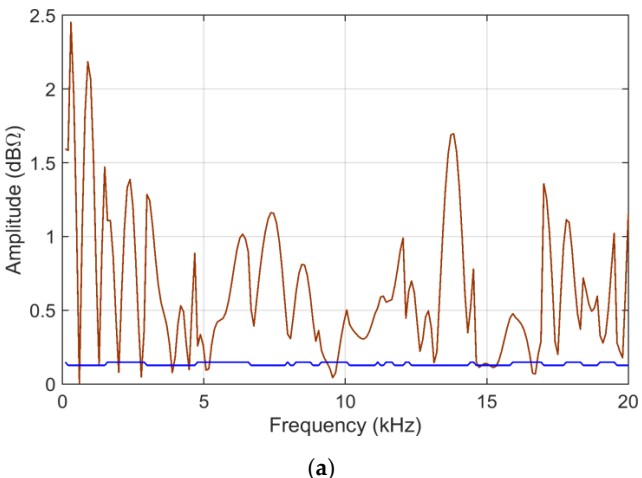

(**a**)

**Figure 9.** *Cont.*

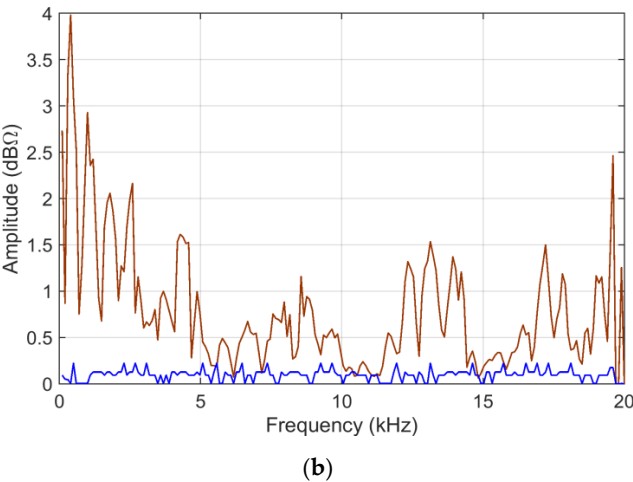

**(b)**

**Figure 9.** $Z$ (47.5 km) and $Z$ (48.0 km) input data in logarithmic scale: (**a**) $ADM_i$ and (**b**) $FDM_i$ sensitivity curves, comparing analytical first-order derivative calculation (blue) and normalized standard deviation of MCM output (brown).

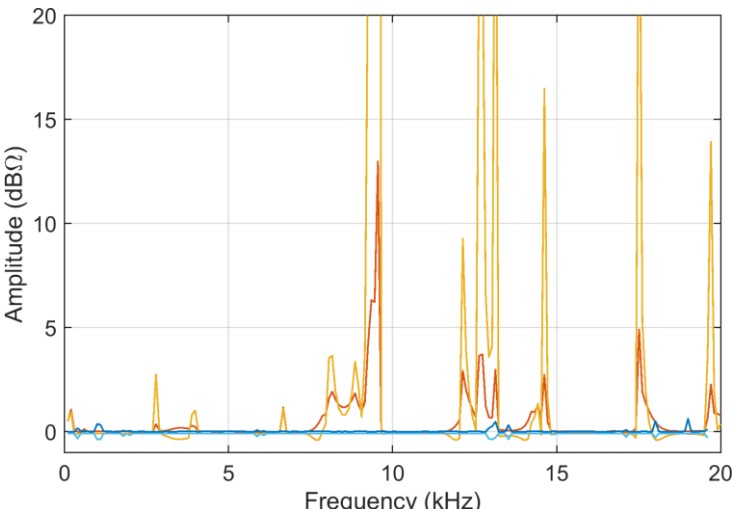

**Figure 10.** $Z$ (47.5 km) and $Z$ (48.0 km) input data in logarithmic scale: skewness in dark color and kurtosis (offset around 0) in light color for $ADM_i$ (orange *sk* and yellow *ku*) and $FDM_i$ (blue *sk* and light blue *ku*).

$ADM_i$ and $FDM_i$ are in general smaller and more evenly distributed, indicating that there are no specific remarkable major errors between the two curves at 47.5 and 48.0 km, but that the difference is spread over the frequency range.

The error between the respective $ADM_i$ and $FDM_i$ estimates (as from direct FSV method application and as a mean of the MCM output) is in general very low, except for a few short intervals amounting to a fraction of %, and in one case, to more than 2%. There is no particular reason for this, except that all these intervals with larger-than-average errors occur for extremely small values of $ADM_i$ and $FDM_i$, namely amplifying unavoidable small differences.

The *sk* and *ku* curves of Figure 10 confirm that more similar curves have a larger amount of outliers (larger kurtosis) that correspond to the smallest $ADM_i$ and $FDM_i$ values (see Figure 9, slightly before 10 kHz, at about 13 kHz and 15 kHz, and then at 20 kHz); however, the scenario is more complex than this, as some medium-value intervals of $ADM_i$ and $FDM_i$ are also characterized by large kurtosis, as for the samples at around 17 kHz.

### 6.2. Test Case 2: Scattering Parameters

$S_{11}$ curves are characterized by broad humps and deep downward peaks (see Figure 11). The two curves refer to a very similar configuration so that they have only one interval in the middle of the frequency axis, where they differ; other small differences are hardly visible. The curves, contrarily to those of railway impedance, proceed from negative values to zero with a smaller overall variation; they are also characterized by some deep dips.

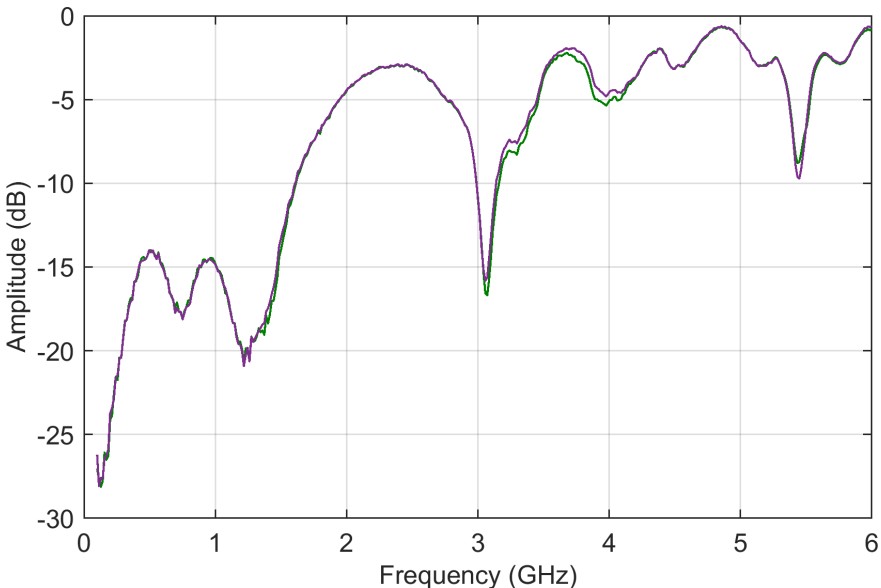

**Figure 11.** $S_{11}$ input data curves 1A and 1B in logarithmic scale (dB) of test case 2.

$ADM_i$ and $FDM_i$ amplitudes for analytical and MCM calculations are shown in Figure 12, accompanied by an estimate of the relative amplitude error.

Observing the reported error (between the direct FSV method application and the mean of the MCM output) there are points with much larger errors; again, these are all in correspondence with very low curve values and small unavoidable differences between the two curves.

Regarding the behaviors of the $ADM_i$ and $FDM_i$ curves, they are positioned in a range of values well below unity (due to the logarithmic scale and the similarity of the two selected curves). The two curves are selected so that they have a localized smaller difference in amplitude along the upward slope between 1.2 and 1.5 GHz, and a larger one of between about 3.2 and 4 GHz; then, they feature a difference of amplitude in the two downward peaks at 3 and 5.5 GHz. In addition, the downward peaks are very steep.

As a result, $ADM_i$ detects well the three intervals with differences in amplitude, although smaller values could have been expected for the first interval between 1.2 and 1.5 GHz; the $ADM_i$ resulting amplitude is instead similar if it is compared to the interval of between 3.2 and 4 GHz. The reason for this is that the larger amplitude difference in the latter is compensated for by a smaller local amplitude in the sense of absolute value, although we know that a −5 dB value is in reality physically larger than a −15 dB one. This confirms the scale problems of the FSV internal indexes.

Figure 13 reports the $ADM_i$ and $FDM_i$ sensitivity curves for the $S_{11}$ scattering parameters case, and Figure 14 shows *sk* and *ku* for $ADM_i$ and $FDM_i$.

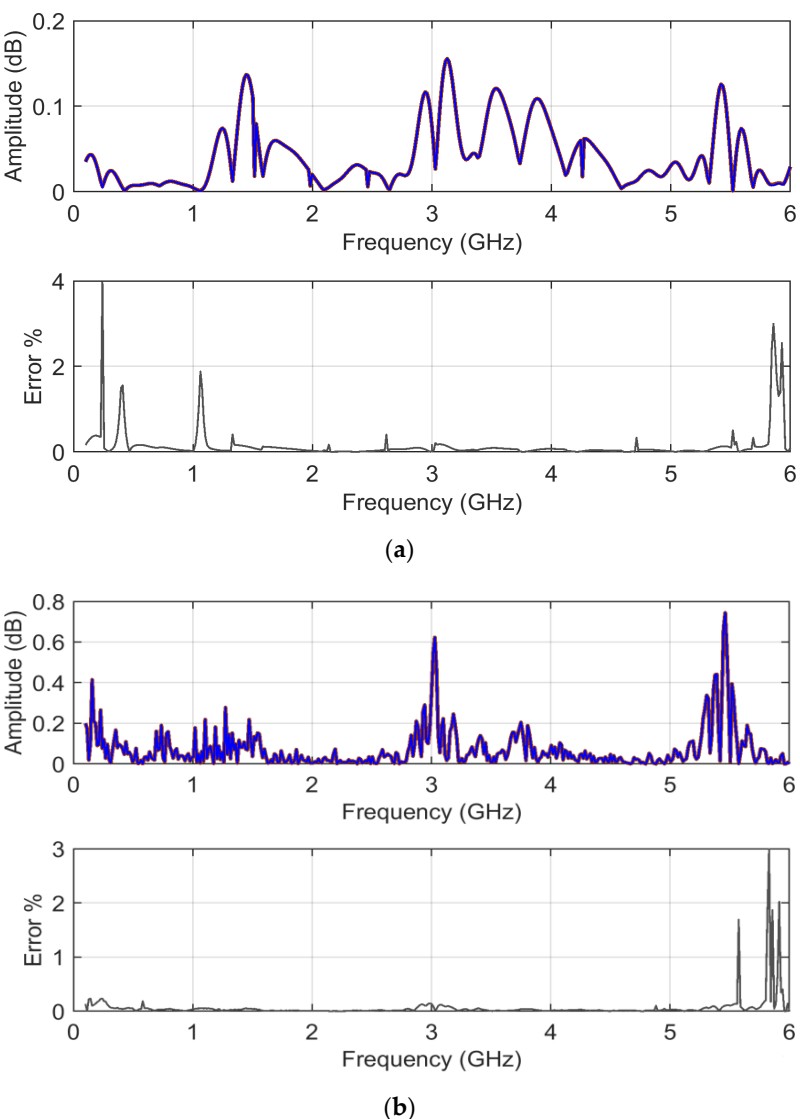

(a)

(b)

**Figure 12.** $S_{11}$ input data in logarithmic scale: (**a**) $ADM_i$ and (**b**) $FDM_i$ plots as provided by analytical FSV calculation (blue) and mean of MCM output (brown); blue and brown curves are overlapped with underlying brown curve visible at the edges of the blue one.

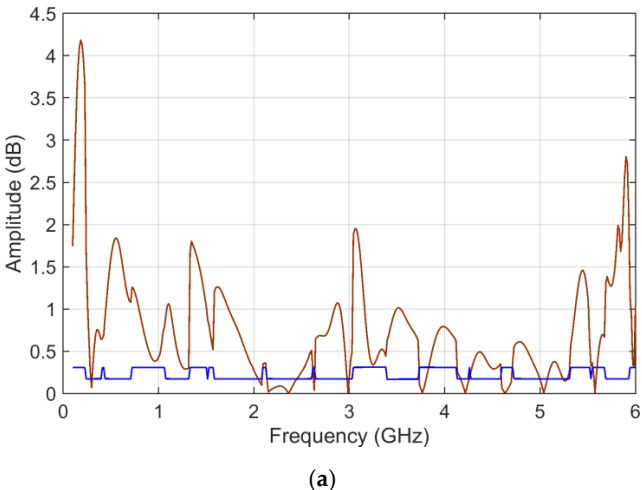

(a)

**Figure 13.** *Cont.*

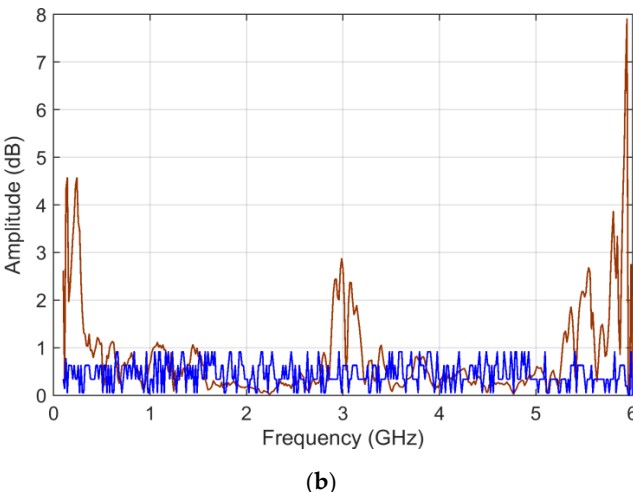

**(b)**

**Figure 13.** $S_{11}$ input data in logarithmic scale: (**a**) $ADM_i$ and (**b**) $FDM_i$ sensitivity curves, comparing analytical first-order derivative calculation (blue) and normalized standard deviation of MCM output (brown).

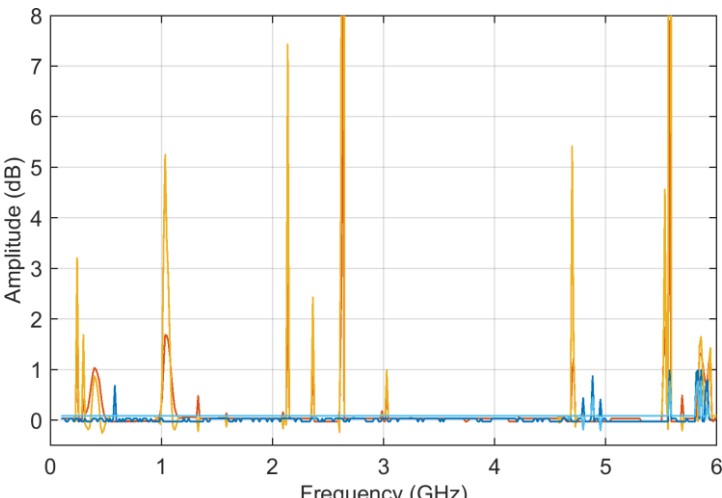

**Figure 14.** $S_{11}$ input data in logarithmic scale: skewness in dark color and kurtosis (offset around 0) in light color for $ADM_i$ (orange $sk$ and yellow $ku$) and $FDM_i$ (blue $sk$ and light blue $ku$).

In the case of scattering parameters with a more limited dynamic range, the sensitivity coefficients that are calculated analytically and that are determined by MCM more or less correspond, showing that the compression of the dynamic range of input data is beneficial. Violations of the normality assumption for the probability distribution of MCM output are several, although limited, clearly corresponding to frequency intervals where the compared data are very similar and where $ADM_i$ and $FDM_i$ have the lowest values.

### 6.3. Normalization of MCM Calculated Sensitivity

Two facts emerge from the results obtained so far:

- The analytically calculated sensitivity using first-order derivatives is always smaller (or much smaller) than the MCM result, proving that the problem is not suitable for the propagation of uncertainty by means of derivatives;
- Comparing linearly and logarithmically scaled data, the differences indicate a second issue for the determination of FSV sensitivity, that of a possible dependency on data format.

The reason for the unsuitability of the analytical approach is for sure that the $ADM_i$ and $FDM_i$ functions are highly non-linear, as demonstrated by the form of the first-order derivatives themselves (where the sign functions are visible and frequent stepwise changes occur). This is caused at least by the use of absolute value operators in the formulation of $ADM_i$ and $FDM_i$.

It is also observed that attempting to increase the order of the calculated derivatives to improve the approximation is unsuccessful, as further derivation of $S_\gamma$ and $S_\theta$ provides a second-order derivative with terms of similar magnitude with the addition of a $1/(2N)^2$ term: only the latter has reduced magnitude, showing that higher-order terms of the Taylor expansion are non vanishing, and that convergence is troublesome.

It is remembered only that the FSV method was first conceived in the field of electromagnetic compatibility, where the data are almost always provided in dB scale, leading us to think that the dB representation is possibly favored.

Comparing the sensitivity values obtained with the scattering parameters (small negative values approaching zero) and railway line impedance values (large values ranging approximately between 10 and several thousands), it is evident that sensitivity depends on the scale of the provided data, and is different for different portions of each data curve. In an attempt to generalize the MCM calculated sensitivity, a sort of normalization was investigated, to decide the amount of sensitivity that one could expect, irrespective of the data magnitude and their range of variation (dynamic scale).

An attempt has been made to normalize $S_{ADMi}$ and $S_{FDMi}$, so as to have a better behavior where their value is scaled by the data magnitude. Since the data have a large dynamic range and a few points may reach very small or large values, causing issues of weird amplification or the attenuation of sensitivity coefficients after division, a local smoothing of the data was used instead. Smoothing is achieved by applying a median filter $mf(m_i,r)$ with a range $r$, which for the reported results was chosen as $r = 11$.

$$\widetilde{S}_{ADM_i} = \frac{S_{ADM_i}}{\mathrm{mf}\,(m_i,r)} \qquad \widetilde{S}_{FDM_i} = \frac{S_{FDM_i}}{\mathrm{mf}\,(m_i,r)} \tag{27}$$

The result is shown below in Figure 15 for the data of test case 1 (railway impedance), represented both in linear and logarithmic scale. It is observed that the normalized coefficients $\widetilde{S}_{ADM_i}$ and $\widetilde{S}_{FDM_i}$ are unitless.

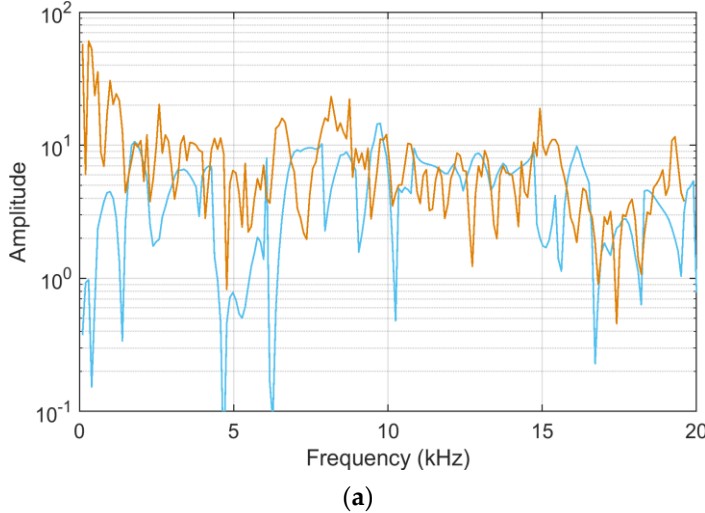

(a)

**Figure 15.** *Cont.*

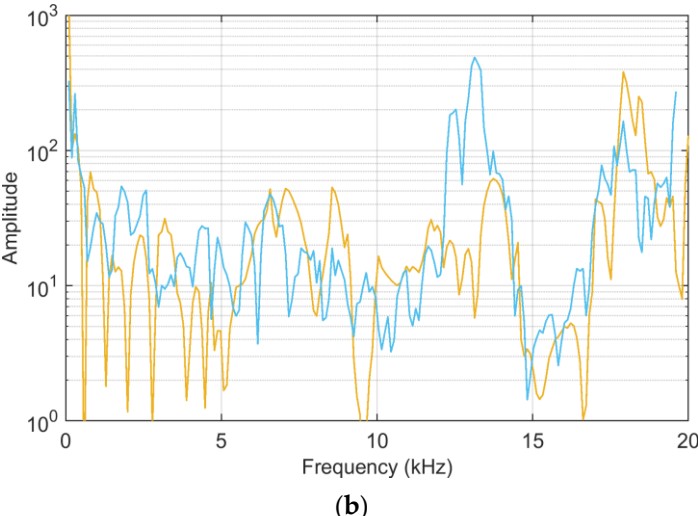

(**b**)

**Figure 15.** Normalized sensitivity coefficients for $Z(45.0$ km$)$ and $Z(45.5$ km$)$ with input data provided in (**a**) linear and (**b**) logarithmic scale: $S_{ADMi}$ is shown in light brown and $S_{FDMi}$ in light blue.

The normalized sensitivity curves do not have a more uniform profile than the sensitivity coefficients curves alone, nor was this the main objective. What the $\widetilde{S}_{ADM_i}$ and $\widetilde{S}_{FDM_i}$ achieve is weighting sensitivity with respect to data sample intensity, and this is relevant, as it is possible to assume that data errors depend on scale, with a larger uncertainty at lower values, but smaller error values. In other words, the normalized sensitivity coefficients spot out areas of small data values with large sensitivity, and at the same time, reduce the influence of large data values.

For completeness, the normalized coefficients are also calculated for test case 2 ($S_{11}$ scattering parameters), showing a moderate range of values, as commented before (see Figure 16). In this case (and similarly for Figure 15b above), although the quantities are in dB, the normalization is intended as a ratio, operated to scale with respect to the numeric values, so that the resulting normalized sensitivity coefficient is unitless and in linear scale.

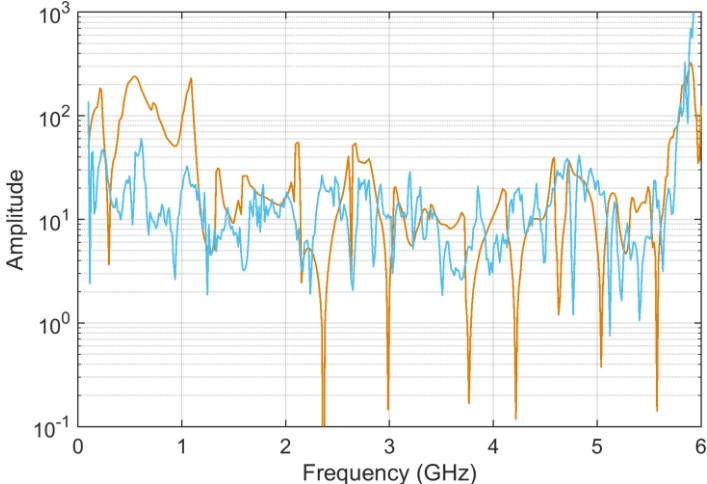

**Figure 16.** Normalized sensitivity coefficients for $S_{11}$ curves in logarithmic scale: $S_{ADMi}$ is shown in light brown and $S_{FDMi}$ in light blue.

We may comment that the $S_{FDMi}$ coefficient suffers from edge effects, increasing always towards the first and last samples of the data vector, as a consequence of the way in which $FDM_i$ is calculated (with the local derivative taking some samples).

Although the sensitivity values $S_{ADMi}$ and $S_{FDMi}$ have been observed in the range of 0.1 to 5 for all data vectors, when normalized by the median filter estimate of the local data

amplitude, the resulting $\widetilde{S}_{ADM_i}$ and $\widetilde{S}_{FDM_i}$ values lie between approximately a few units (or 10) and some hundreds of units.

## 7. Conclusions

FSV is a relevant algorithm for the validation of simulation results, and in general, to assess the similarity between the simulation and/or experimental data. It belongs to a family of algorithms to spot out similarity, beyond a simple comparison between arrays of values, evaluating broader characteristics such as slope, number of peaks, and so on. FSV has proven to be effective in several engineering fields, focusing on electromagnetism and electromagnetic compatibility.

This work has addressed the characterization of FSV sensitivity, as is recommended for any instrument (measuring tool), with the task of providing a similarity assessment of the said simulation and/or experimental data. The commonly adopted approach is that of determining sensitivity coefficients by first-order derivatives (in case some assumptions regarding Taylor approximation and curve behavior hold), and more generally, by Monte Carlo method (MCM) simulations.

We have demonstrated that, due to the nature of the FSV indexes ($ADM_i$ and $FDM_i$), the straightforward calculation of first-order derivatives leads to inaccurate, and sometimes inconsistent, results. The first-order derivative results are always much lower if compared to MCM (up to two orders of magnitude) and they highly depend on the type of data and whether they are linearly or logarithmically projected. Logarithmic compression (or another scale, compression mapping) should be used in order to have a better distribution of sensitivity of $ADM_i$ and $FDM_i$; this ensures more similarity between the sensitivity coefficients resulting from first-order derivatives, and the dispersion of MCM output (differences are reduced to a factor of 1 to 5, on average), showing that FSV non-linearity is reduced. MCM always delivers consistent results and has been considered as the reference to evaluate FSV sensitivity and uncertainty. MCM is an approach endorsed by GUM, and should be preferred for a quantitative analysis of uncertainty propagation in FSV expressions (and in general, for complex systems and algorithms).

By calculating the skewness and kurtosis of the MCM sample set, indexes are demonstrated to have symmetric, nearly normal distributions; in some cases, a few samples, or narrow intervals, deviate from normality and exhibit a more relevant number of outliers, with kurtosis being several times larger than Gaussian kurtosis (equal to 3). This occurs when the $ADM_i$ and $FDM_i$ values are extremely small (the compared curves are quite similar locally) and the spread of values beyond normality (long tails) is in line with the observed underestimation of the Excellent interval [9,20]. This supports a derivation of uncertainty and confidence of interval based on a confirmed Gaussian assumption.

The derivation of uncertainty information by MCM is problem specific, although general sensitivity figures were derived, showing multiplicative values starting from a few % to some units, namely indicating that the amplification of data noise (or uncertainty) occurs only at some intervals. To obviate scale problems, normalization of the sensitivity coefficients was carried out using a smoothed version of the original data: the objective was to provide a more fair representation of sensitivity, as its strong dependency on the local $ADM_i$ and $FDM_i$ values was consistently observed. The resulting normalized sensitivity ranges approximately between about 10 and 100, with narrow downward peaks (irrelevant) and some larger values, especially at the beginning and end of data vectors (the edge effect, caused by the $FDM_i$ calculation method).

The effectiveness of FSV as a gauging tool or instrument must be evaluated by means of the propagation of uncertainty and evaluation of sensitivity to noisy data and data uncertainty. However, no better validation metrics are known, from the point of view of accuracy and limited sensitivity to data.

Further developments and studies are in the direction of evaluating the overall uncertainty of the grade and spread of FSV output with methods that are more suitable for

studying histograms and clustering variability, as well as some of the FSV variants that have more recently been proposed.

**Author Contributions:** The contribution of the two authors toward conceptualization, methodology, software, experimental validation, and writing is the same. All authors have read and agreed to the published version of the manuscript.

**Funding:** This research received no external funding.

**Institutional Review Board Statement:** Not applicable.

**Informed Consent Statement:** Not applicable.

**Conflicts of Interest:** The authors declare no conflict of interest.

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
