# Peer review of "Uncertainty and Sensitivity of the Feature Selective Validation (FSV) Method"

_electronics, doi:10.3390/electronics11162532_

Round 1
Reviewer 1 Report
I have read with interest the manuscript titled "Uncertainty and sensitivity of the Feature Selective Validation (FSV) Method", submitted to Electronics. The Authors present a mathematical application of the FSV method based on the IEEE Std.1597.1.
I believe the present study is of potential interest for the readers of this Journal. However, before being adequate for publication, the Authors are invited to clarify the points listed below.
1) In Section 1 (page 1), the Authors discuss the fact that "models can replace the physical system". Since this topic is quite wide, the Authors should add more References, e.g. Refs. [4-7] seems to be not enough.
Furthermore, Ref. [1] is not relevant in this Section, thus the Authors are invited to remove it (and to avoid useless self-citations!).
2) In Section 1 (page 1), the Authors could mention that another important reason for using modelling and simulation approaches is that models are usually cheaper and faster than running experiments. Also, a mention to the modern "digital twin" technologies would be beneficial for the completeness of the Section.
3) In Section 1 (page 2), the Authors mention the "Type B approach", which is not clear to the reader(s). The distinction between Type A and Type B methods, indeed, will be clear only later in the paper.
The Authors are invited to rephrase the paragraph and thus to provide a more clear flow of information.
4) In Section 2 (page 3), the Authors write "derivates are applied to the original vectors with their sign, not their moduli". This statement is quite trivial and this specification can strongly confuse the reader. Probably the Authors refer to old errors and/or misunderstanding related to their previous work. Therefore, the Authors are invited to remove this point or to rephrase it to make it more clear.
5) In Section 2 (page 3), Equation (1) seems to report wrong values of the index. One can read the intervals X[0...4], X[5...Ib] and X[Ib...N-1]. Maybe the last interval is wrong and it should be X[Ib+1...N-1]. The Authors are invited to review this point.
6) In Section 2 (page 4), Equation (10) reports a numerical coefficient equal to 7.2. Since it is not an integer number, it is not clear to the reader(s) where this coefficient come from. The Authors are invited to review this point or to add some References.
7) In Section 2 (page 4), Equation (12) seems to be useless for the comprehension of the manuscript, since the quantity FDM is not used anymore later. The Authors should consider of removing this Equation.
8) The Authors are invited to provide References for every formula they are introducing, otherwise it seems that they are an original output of this manuscript (it may not be the case). Specifically, one or more References are welcome for Equations (5-12), Equations (13,14) and Equations (16-26).
9) In Section 5 (page 7), the Authors are invited to provide at least one classical Reference for the Monte Carlo method.
10) In Section 6 (from page 14), it is not clear why Figure 8a, as well as Figure 12, present so large values of the error, especially at low frequencies. The Authors are invited to clarify this point in the manuscript.
11) In Section 6 (page 20), it is not clear where is a linear scale in Figure 15a.
In addition, the Authors are warmly invited to revise the minor points listed below.
- Title (page 1). Maybe it is better to write "Method" with non-capital letter, since it is not part of the acronym "FSV".
- Section 1 (page 1). Please substitute "numeric models" with "numerical models".
- Section 1 (page 1). Please substitute "a simple metric that measure" with "a simple metric that measures [...]".
- Section 3 (page 5). Please substitute "are those considered in 11" with "are those considered in [11]".
- Section 3 (page 5). Please remove the empty brackets "( )" after mentioning the function F (2 times in the page).
- Section 3 (page 5). Please susbtitute "GUM [10], Clause G.2.1" with "GUM Clause G.2.1 [10]", for consistency with the previous entry.
- Section 3 (page 5). Please remove the single bracket "(" before "this will be verified indirectly".
- Section 6 (page 9). Please add a Reference in the caption of Figure 1, if not original product of this work.
- Section 6 (page 9). In Figure 1, a colour legend inside the caption is forbidden: please insert a colour legend in the plot.
- Section 6 (page 9). In the caption of Figure 1, please substitute "to note the closer" with "note the close".
- Section 6 (from page 9). Please use a consistent unit of measure for the Amplitude in Figures 1-16, because many figures have a symbol Omega, other have nothing after amplitude, and other have dB. Please clarify.
- Section 6 (page 10). The brown color mentioned in the caption of Figure 2 is not visible.
- Section 6 (from page 10). The brown color mentioned in the caption of Figure 3 is not visible.
- Section 6 (page 14). The brown color mentioned in the caption of Figure 8 is not visible.
- Section 6 (from page 16). The brown color mentioned in the caption of Figure 12 is not visible.
- Section 6 (page 19). Please remove the empty brackets "( )" after mentioning the function mf.
- Section 6 (page 19). Please substitute "for the reported results was chosen equal to 11" with "for the reported results was chosen equal to [11]", if really referring to Ref. [11].
- Section 7 (page 21). Please substitute the section title "5. Conclusions" with "7. Conclusions".
- Section 7 (page 21). Please substitute "Gaussian kurtosis (=3)" with "a Gaussian kurtosis (equal to 3)".
Author Response
Dear Reviewer, thank you for your extensive and articulated review.
We have provided replies and indications of the amendments to the manuscript. We hope that they satisfy your remarks.
Please, see attached file.

Reviewer 2 Report
Acronyms DFT and IDFT appear first on page 3, and there is no description of their meaning.
In page 7, when the "Latin hypercube sampling" is mentioned, the following new reference [19] related to this samping method should be inserted:
[19] Xiangqi Li, Yunfeng Li, Li Liu, Weiyu Wang, Yong Li, and Yijia Cao. Latin Hypercube Sampling Method for Location Selection of Multi-Infeed HVDC System Terminal. Energies 2020, 13(7), 1646.
In Figure 1, include a legend indicating line colors for the several positions between 45 km and 48.5 km, and remove those distances from Figure 1 description, that could then be: Line impedance curves of test case 1 for several positions along the line. To note similarity of curves in the last positions beyond 47 km.
Include the unit (Ohm) for the amplitude quantity in the vertical axis of Figure 4, Figure 5, and Figure 9.
It seems that the plots of Figure 12 are not in a logaritmic scale. If that is the case, please change the units of amplitude in this figure to (Ohm).
Include the units for amplitude in the vertical axis of the plots in Figure 13. It seems that these are not on a logaritmic scale.
Include the unit (DB) for the amplitude quantity in the vertical axis of Figure 15.
It seems that, plots of figures 2 to 7 do not include data for one of the impedances Z(45 km) or Z(45.5 km). Simillarly it seems that, plots of figures 8 to 10 do not include data for one of the impedances Z(47.5 km) or Z(48 km)
The document should be improved with relation to the plots of Figures, to clarify these, namely:
1) No legends are included in figures and descriptions are extensive and confusing;
2) There are linear scales where should be logaritmic scales. Sometimes Amplitude units are not used properly according to the adopted scale.
Conclusions can be improved with representative results from comparing the FSV with MCM.
Author Response

(The authors gave the same response as above.)

Reviewer 3 Report
There is no synchronization in the title as well, i.e., s for the term “sensitivity” should be capitalized in the given title “Uncertainty and sensitivity of the Feature Selective Validation (FSV) Method”. Secondly, it is not a good idea to abbreviate the terms in the title.
Please abbreviate the terms at their first occurrence, i.e., FSV and GUM in the abstract. And, similarly in the introduction section. Further, the abstract is not up to mark, it should be written again and please add some numerical results at the end of the abstract.
Usually, if you want to write “e.g” then the correct way to write it is “, e.g., then your sentence”. Please correct the kind of mistake in the whole manuscript.
The contributions of the paper are not clear. Please mention them clearly at the end of the introduction section. Add a separate literature review section in the paper with a tabular form describing the proposed schemes, their limitations, and main features.
And there are many grammatical and typo mistakes thought the paper. The following work needs serious English revisions to ease the reader.
Why the following research is carried out and its importance is not clear. Please make it clear in the abstract and in the introduction as well.
No equation is referred in the paper, please refer to the ones that do not belong to you. Further, to write the intervals the correct way is as follows A= [1,2, …, 5]. Please update your paper’s equations accordingly. Further, take care of spaces in the equations, i.e., eq (4) absolute signs are not readable.
Please add a subsection with problem formulation and discuss the proposed solution. For the time being, both sections are missing. Then, add the complexity analysis of the proposed solution as well.
Simulation and results are not well presented and explained. Further, comparison analysis with the benchmark scheme is missing and paper organization is very weak which makes the paper difficult to follow.
You can easily merge the plots in Figures 2 and 3. I mean without logarithmic pre‐processing and with logarithmic pre‐processing for both ADM and FDM. A similar case can opt for figures 4 and 5 for analytical calculations and MCM simulation and results. Further, proper reasoning is missing from these plots. Consider the above comment for other similar plots too.
Please update the conclusion properly and precisely and in the end, add some recent references to the paper.
Once the authors incorporate these comments then I have no objection to accept the following work. Overall authors have done good work.
Author Response

(The authors gave the same response as above.)

Round 2
Reviewer 1 Report
Thank you for having answered to all my questions and improved the quality of the submission.
I can now recommend publication of the manuscript, as soon as you take care of the following minor points.
1) Please substitute "wind parks" with "wind farms" in the Introduction.
2) Sorry if I ask again, but the units of the Amplitude in EVERY FIGURE are different: Omega - dB - dBOmega - nothing. Can you please make them all consistent?
Reviewer 3 Report
Authors have significantly improved the manuscript now. I have no further comments.